# High Dose Fish Oil Added to Various Lipid Emulsions Normalizes Superoxide Dismutase 1 Activity in Home Parenteral Nutrition Patients

**DOI:** 10.3390/nu16040485

**Published:** 2024-02-08

**Authors:** Stanislav Sevela, Eva Meisnerova, Marek Vecka, Lucie Vavrova, Jana Rychlikova, Martin Lenicek, Libor Vitek, Olga Novakova, Frantisek Novak

**Affiliations:** 14th Department of Internal Medicine, 1st Faculty of Medicine and General University Hospital, Charles University, 128 08 Prague, Czech Republiceva.meisnerova@vfn.cz (E.M.); marek.vecka@lf1.cuni.cz (M.V.); vavrova3@seznam.cz (L.V.); jana.rychlikova@lf1.cuni.cz (J.R.); libor.vitek@lf1.cuni.cz (L.V.); 2Institute of Medical Biochemistry and Laboratory Diagnostics, 1st Faculty of Medicine and General University Hospital, Charles University, 128 08 Prague, Czech Republic; martin.lenicek@lf1.cuni.cz; 3Department of Physiology, Faculty of Science, Charles University, 128 00 Prague, Czech Republic; olnov@natur.cuni.cz; 4Institute of Physiology, Academy of Sciences of the Czech Republic, 142 20 Prague, Czech Republic

**Keywords:** chronic intestinal failure, short bowel syndrome, oxidative stress, fibroblast growth factor 19, liver function tests, bile acids

## Abstract

(1) Objectives*:* Intestinal failure in home parenteral nutrition patients (HPNPs) results in oxidative stress and liver damage. This study investigated how a high dose of fish oil (FO) added to various lipid emulsions influences antioxidant status and liver function markers in HPNPs. (2) Methods: Twelve HPNPs receiving Smoflipid for at least 3 months were given FO (Omegaven) for a further 4 weeks. Then, the patients were randomized to subsequently receive Lipoplus and ClinOleic for 6 weeks or vice versa plus 4 weeks of Omegaven after each cycle in a crossover design. Twelve age- and sex-matched healthy controls (HCs) were included. (3) Results: Superoxide dismutase (SOD1) activity and oxidized-low-density lipoprotein concentration were higher in all baseline HPN regimens compared to HCs. The Omegaven lowered SOD1 compared to baseline regimens and thus normalized it toward HCs. Lower paraoxonase 1 activity and fibroblast growth factor 19 (FGF19) concentration and, on the converse, higher alkaline phosphatase activity and cholesten concentration were observed in all baseline regimens compared to HCs. A close correlation was observed between FGF19 and SOD1 in baseline regimens. (4) Conclusions: An escalated dose of FO normalized SOD1 activity in HPNPs toward that of HCs. Bile acid metabolism was altered in HPNPs without signs of significant cholestasis and not affected by Omegaven.

## 1. Introduction

Intestinal failure (IF) is a state of insufficient intestinal capacity to fulfil nutritional demands that most often occurs because of short bowel syndrome. Patients suffering from IF depend on long-term home parenteral nutrition (HPN) [1]. Various lipid emulsions (LEs) are essential components of HPN. Not only are the triacylglycerols contained in LEs a source of energy, the fatty acids (FAs) bound in them are incorporated into the cell membrane lipids of the human body and, either by themselves or as precursors of signaling molecules, are significant modulators of the immune response and inflammation [2,3]. FA composition of LEs also plays an important role in regulating the production of reactive oxygen species (ROS) that accompany IF [4,5]. Cu/Zn superoxide dismutase (SOD1) is believed to play a major role in the first line of antioxidant defense by catalyzing the dismutation of superoxide anion radicals to form hydrogen peroxide and molecular oxygen [6].

The combination of compromised bowel function with adverse effects of HPN, resulting in varying degrees of liver damage, is termed IF-associated liver damage (IFALD). This condition is well described, especially in pediatric HPN patients [7]. Although technical improvement and development of new parenteral nutrition formulas have reduced the incidence and severity of IFALD, it remains a major cause of morbidity and mortality in patients on long-term HPN [8,9]. In adults, IFALD is defined as an elevation of liver enzymes to two or three times above the upper limit of normal that occurs at least 6 months after starting on HPN [9,10]. Other factors have also been described in association with IFALD, such as disruption of bile acid (BA) reabsorption and synthesis of fibroblast growth factor 19 (FGF19). This factor is produced in the ileum following the absorption of BAs from the intestinal lumen [11,12]. BAs regulate their own synthesis in the liver by binding to the farnesoid X receptor in the distal ileum, which induces the production of FGF19. After secretion, FGF19 reaches the liver via the portal circulation, where it provides negative feedback control of BA synthesis. FGF19 and BAs are also considered to contribute to the control of inflammation and oxidative stress (OS) [9,11,13].

IF is often associated with microbial imbalance and dysbiosis in the gut, sometimes manifesting as small intestinal bacterial overgrowth. This condition leads to changes in intestinal mucosal permeability. Decreased enteral intake is another important factor contributing to gut epithelial barrier leakage [9,14,15]. Endotoxemia resulting from lipopolysaccharide (LPS) leads to low-grade inflammation, which probably contributes to liver damage [14,16]. These conditions produce an inflammatory response, which, among other effects, increases OS. Overall, IF can be considered an inflammatory state associated with an antioxidative system imbalance [14]. Some studies suggest that omega-3 polyunsaturated fatty acids (*n*-3 PUFAs) can ameliorate this mucosal inflammation and damage [17,18]. Current clinical practice offers a range of lipid emulsions encompassing a rather diverse spectrum of FAs and ratios of monounsaturated acids (MUFA)/PUFA and *n*-6/*n*-3 PUFA.

The effect of *n*-3 PUFAs on IFALD has been studied and amelioration of inflammation has been suggested as one of the possible mechanisms producing that improvement [19]. Fish oil (FO) is rich in *n*-3 PUFAs and α-tocopherol and it contains no phytosterols. Many studies have provided evidence that *n*-3 PUFAs can also improve liver function [3]. The main mechanism is still unclear and may include increasing antioxidant activity or decreasing the inflammatory response. However, *n*-3 PUFAs are highly susceptible to oxidation due to their multiple double bonds. Lipid oxidation can induce cell damage and stimulate the inflammatory response. Thus, it is unclear whether *n*-3 PUFAs act as anti-oxidative or pro-oxidative agents [2,3,19]. This study follows on our previously published clinical trial comparing two FO-enriched LEs, Smoflipid with natural FO and Lipoplus with re-esterified FO, with ClinOleic, an LE with low PUFAs and high MUFAs [2]. Using Omegaven, this study tested additional supplementation of FO in all three baseline LEs and demonstrated the suppression of in vitro LPS-stimulated cytokine production by escalated dose of FO. Here, in this sub-study, we present the effect of *n*-3 PUFAs upon antioxidant status, BA and FGF19 concentrations, and the liver enzymes across a single set of home parenteral nutrition patients (HPNPs).

## 2. Material and Methods

### 2.1. Design and Participants

This crossover-controlled clinical study involved 12 patients with long-term HPN. Inclusion criteria were as follows: age > 18 years, parenteral nutrition administration > 4 days/week with expected duration > 8 months, and stable clinical condition without complications in the past 2 months. The exclusion criteria were the following: active cancer or its treatment, established immunodeficiency, and advanced organ dysfunction from chronic disease. All consecutive HPNPs (55 patients) between January 2012 and July 2016 were screened for eligibility to be included in the study. Every eligible HPNP was assigned to three cycles of isocaloric HPN using different commercially available lipid emulsions: Smoflipid™ (Fresenius Kabi, Bad Homburg, Germany; 30% soybean oil [SO], 30% medium chain triglycerides [MCT], 25% olive oil [OO], 15% FO), Lipoplus™ (BBraun, Melsungen, Germany; 40% SO, 50% MCT, 10% FO), and ClinOleic™ (Baxter, Deerfield, IL, USA; 20% SO, 80% OO). After at least 12 weeks of Smoflipid, Omegaven™ (Fresenius Kabi, Bad Homburg, Germany; containing 10% FO) was added for a further 4 weeks. After this cycle, patients were randomized to receive subsequently two cycles with Lipoplus and ClinOleic for 6 weeks plus 4 weeks of added Omegaven after each cycle in a crossover design. Fasting blood samples (>5 h without parenteral nutrition) were taken for subsequent analyses after each cycle (Figure 1). Parenteral nutrition was administered in individually prepared bags at the hospital pharmacy. All formula components including daily vitamin and trace element supplements were used in accordance with the European Summary of Product Characteristics. LE compositions of individual oils, egg phospholipids, vitamin E (all patients were monitored at least twice a year to maintain normal levels of vitamin E), and FO dosing, including a description of their analyses, were presented in a previous publication [2]. Healthy controls (HCs) were matched according to age (±5 years) and sex (Table 1).

### 2.2. FGF-19, Cholesten, and Bile Acid Measurement

Serum concentrations of FGF19 were determined by enzyme-linked immunosorbent assay (FGF19 Quantikine ELISA kit, R&D Systems, Minneapolis, MN, USA). Cholesten (C4) and BAs were measured using liquid chromatography–mass spectrometry after acetonitrile precipitation as described previously [20].

### 2.3. Antioxidant Enzymes

The activities of antioxidant enzymes were measured in erythrocytes spectrophotometrically using kinetic methods described previously [21,22]. All routine biochemical tests were performed at the Institute for Medical Biochemistry and Laboratory Diagnostics of General University Hospital in Prague.

### 2.4. Fatty Acid Analyses

Plasma phospholipids (PPLs) were extracted according to a modified method of Folch et al. [23]. The PPLs were separated by one-dimensional thin-layer chromatography and prepared FA methyl esters were separated and detected by gas chromatography [2,24].

### 2.5. Statistical Analyses

Power analysis was performed using relevant variability data for main FA classes and assuming a two-sided 5% significance level. The sample size of 12 was considered sufficient to ensure that the power of one-way analysis of variance (ANOVA) was 80% and able to detect differences in the mean FA proportions [25]. For parametric analyses, differences between HPNPs and HCs were evaluated using one-way ANOVA with Newman–Keuls post hoc test comparisons. For nonparametric analyses, the Kruskal–Wallis test was used. For dependent analyses of differences between baseline HPNPs and between the baseline and +Omegaven treatments, the paired sample *t*-test was used. For a nonparametric comparison of baseline and +Omegaven treatments, the Wilcoxon signed-rank test was used. Spearman correlation coefficients were calculated for associations between FGF19 and SOD1, alkaline phosphatase (ALP) and C4, and ALP and BAs. All statistical analyses were performed with StatSoft Statistica 12 CZ software. Charts and grids were made in MS Office 365 Excel. The cutoff for statistical significance was set at *p* < 0.05 with Bonferroni correction in multiple comparisons.

## 3. Results

### 3.1. Population Characteristics

Table 1 presents demographic and parenteral nutrition parameters for HPNPs. All of the LEs tested were well tolerated and no clinical complications were reported.

### 3.2. Biochemical Parameters

In all HPN regimens, total cholesterol, HDL cholesterol, Apo A1, Apo B, and calcium concentrations were lower and transferrin higher in comparison with HCs. There were no differences in the concentration of triacylglycerols, LDL cholesterol, serum amyloid A, and lipoprotein (a) (Table 2).

### 3.3. FGF19, Bile Acids, and Liver Function Tests

We found lower FGF19 and conversely higher C4 concentrations in all HPNPs versus the HC group. The addition of Omegaven significantly reduced the C4 concentration only in the Smoflipid regimen and there was a slight upward trend in FGF19 for all regimens. The concentration of total BAs tended to have higher values in all basal regimens versus HCs (Figure 2). The serum concentration of glycine-conjugated deoxycholic acid was significantly lower in all HPNP groups and lithocholic acid was lower only in Smoflipid, Lipoplus, ClinOleic, and ClinOleic + Omegaven regimens compared to HCs. No differences were found in other BA concentrations. There was an increased ratio of primary to secondary BAs in all HPNPs compared to HCs. As for the liver function tests, the total bilirubin concentration, aspartate transferase (AST), alanine transferase (ALT), and γ-glutamyltransferase (GGT) enzyme activities did not differ among regimens and versus HCs while only that of alkaline phosphatase (ALP) increased in all HPNPs. Nevertheless, the values were still within the physiological range. The addition of Omegaven decreased the ALP activity only in the Smoflipid regimen. We also found a significant positive correlation between FGF19 and SOD1 occurring only in basal HPN regimens, disappearing after the addition of Omegaven. Furthermore, ALP was positively correlated with C4 concentrations and with the primary/secondary BA ratio (Table 3).

### 3.4. Antioxidant Enzymes

Activities of antioxidant enzymes in erythrocytes are presented in Figure 3. The most notable results of this study were observed in SOD1 activities. In all basal HPN regimens, the SOD1 values were increased compared to those in the HCs group. Compared to the basal regimens alone, Omegaven administered with basal emulsions significantly lowered SOD1 activities toward those of the HCs, whereas CAT and GPX1 activities in erythrocytes did not differ among HPN regimens or relative to HCs. GR activity was significantly lower only in the Lipoplus + Omegaven compared to the basal Lipoplus regimen. PON1 activity was lower in all HPN regimens compared to HCs. PON1 concentration generally tracked its activity values but the difference between HPNPs and HCs was significant only in the case of Lipoplus. Similarly, HDL cholesterol concentrations were lower in HPN regimens versus HCs. The ratio of oxidized LDL cholesterol to total LDL cholesterol was higher in all HPNPs except in the Lipoplus + Omegaven regimen, where the difference from HCs was not significant.

### 3.5. Fatty Acid Proportion in Plasma Phospholipids

As shown in Table 4, we analyzed the fatty acid profile in PPLs of HPNPs and HCs. The proportions of palmitic (16:0), eicosapentaenoic (20:5*n*-3, EPA), docosapentaenoic (22:5*n*-3), docosahexaenoic (22:6*n*-3, DHA), total saturated fatty acids (SFAs), and *n*-3 PUFAs were elevated and, conversely, linoleic (18:2*n*-6), α-linolenic, dihomo-γ-linoleic (20:3*n*-6), arachidonic acid (20:4*n*-6), total *n*-6 PUFAs, and total PUFAs were reduced in the PPLs for all LE regimens compared to their levels in the HCs. Comparing HPNPs on baseline emulsions, we observed lower dihomo-γ-linoleic acid (20:3*n*-6) and total *n*-6 PUFAs and, conversely, higher 20:5*n*-3 and total *n*-3 PUFAs proportions in the PPLs on the Smoflipid compared to the ClinOleic regimen. There were no differences between the Smoflipid vs. Lipoplus and Lipoplus vs. ClinOleic regimens. Finally, the addition of Omegaven to the basal formulas further reduced the proportion of *n*-6 and increased the proportion of *n*-3 PUFAs in all parenteral interventions.

## 4. Discussion

We recently reported a subset of results from this study regarding the impact of three different LEs with supplemented doses of FO on serum cytokine concentrations and in vitro cytokine production in HPNPs [2]. We concluded that FO-supplemented parenteral nutrition suppresses in vitro LPS-stimulated cytokine production. Moreover, the LPS-stimulated production of IL-6 was negatively correlated with the parenteral dose of EPA + DHA, suggesting an anti-inflammatory effect of *n*-3 PUFAs [2]. In this sub-study, we present changes in the FA profile of PPLs like those published for erythrocyte phospholipids (EPLs) in the aforementioned article [2]. We observed a significantly decreasing effect on arachidonic acid and an increasing effect on EPA and DHA proportions in the PPLs of HPNPs that resulted in the decline of the *n*-6/*n*-3 PUFA ratio, thus confirming a successful incorporation of FAs from Omegaven after all of the baseline LE regimens. The FA profile in EPLs has long been considered a marker of tissue FA status [26].

Speaking about the effects of *n*-3 PUFA supplementation on redox status is always controversial [27]. Most studies confirm that *n*-3 PUFAs, due to their large numbers of double bonds, increase OS, which at the same time stimulates the expression of transcription factors upregulating antioxidant systems [19]. Increased OS is one of the major mechanisms contributing to the pathogenesis of several chronic diseases associated with low-grade inflammation [27,28], as observed also in our HPNPs. This assumption is supported by our finding of higher erythrocyte SOD1 activity and plasma oxidized low-density lipoprotein concentration in all basal HPN regimens compared to HCs. Increasing the intracellular expression of SOD1 may be a highly effective approach to ameliorate cellular injury during increased OS [6]. We recently confirmed increased OS in HPNPs on both FO-supplemented LEs and ClinOleic formulations, with malondialdehyde concentration and SOD1 activity being higher in the FO-supplemented LEs than in the ClinOleic group, suggesting that the *n*-3 PUFA-rich emulsions induce OS [3]. We did not, however, find this difference in the present study. The explanation for this discrepancy may lie in the fact that the 6-week washout period for Omegaven before the subsequent basal emulsion was too short and thus ClinOleic remained contaminated with *n*-3 PUFAs, as evidenced by the PPL FA composition. As for Omegaven, it normalized SOD1 activity in all three regimens nearly to the HC level. The effect of *n*-3 PUFA supplementation on serum SOD1 activity has been studied in 10 trials. A meta-analysis of these trials showed no significant difference in SOD1 serum activity between the *n*-3 PUFAs-supplemented groups compared to those given a placebo regardless of the duration of administration or dose of *n*-3 PUFAs (200–2500 mg EPA+DHA) [27]. We administered much higher doses of FO by adding Omegaven (6300–8400 mg EPA+DHA) in our study and to our knowledge, no comparable intervention in human subjects has yet been published. However, the antioxidant effect of alpha-tocopherol, which was given to the patients together with a high dose of fish oil, should also be considered. There are studies that confirm the protective effect of this vitamin against the peroxidation of PUFAs but in general, the results are quite controversial; many studies also demonstrate its pro-oxidant effect [19]. Miloudi et al. measured hepatic markers of OS in newborn guinea pigs infused for 4 days with emulsion compounded with Intralipid 20% or Omegaven. Pure Omegaven reduced OS and the liver inflammation associated with parenteral nutrition [29]. Interestingly, a relationship between FGF19 and OS has recently been reported [30]. This is consistent with our finding of a close correlation between FGF19 concentration and SOD1 activity in all basal emulsions, which disappeared after Omegaven administration that had normalized SOD1 nearly to HC values.

PON1 is an HDL-associated protein, the function of which is to protect LDL particles from oxidative modifications through its ability to prevent the generation of pro-inflammatory oxidized phospholipids by ROS induced by inflammation [31]. Regarding our HPNPs displaying a low-grade inflammatory state [2], both PON1 activity and concentration were reduced compared to those of HCs. PON1 activity is reduced in inflammatory bowel disease, reflecting its activity and inflammation severity [32]. There is evidence that the amount and composition of dietary lipids are key factors in the modulation of PON1. The molecular mechanisms involved include an effect on PON1 hepatic synthesis or secretion and/or modification of PON1 interactions with HDL. Changes in PON1 activity could also be related to dietary intake of oxidized lipids that behave as PON1 inhibitors [33]. Fuhrman et al. evaluated serum oxidation in mice supplemented with various FAs or oils. DHA (purified fatty acid) and fish oil (type unspecified) enhanced serum oxidation and decreased PON1 activity compared to oleic acid, olive oil, linoleic acid, and soybean oil [34]. The effect has been confirmed also in humans [35]. Our results show that additional *n*-3 PUFA supplementation using Omegaven had no effect, thus suggesting no further escalation of ROS. This result could indicate that the antioxidant system was sufficiently stimulated to handle additional *n*-3 PUFAs prior to the administration of Omegaven and that is shown also by normalized SOD1 activity.

Considering the important role of BAs not only in intestinal but also in hepatic metabolism, we set out to investigate their changes in patients with IF and the possible influence of *n*-3 PUFAs. We found no significant differences in plasma total BA values even with respect to HCs. The ratio of primary to secondary BAs was nevertheless higher in all HPNPs compared to HCs due to lower secondary BAs in the HPNPs. On the other hand, differences in FGF19 and C4 between HPNPs and HCs demonstrate an alteration of BA metabolism in terms of increased synthesis of BAs to cover intestinal losses while the total BA pool is preserved. Overall, our findings are consistent with a previously published study focusing on the spectrum of BAs in short bowel syndrome [36]. Given the heterogeneity of the HPNPs cohort, complexity of the BA metabolome, and relatively low patient numbers, only trends in the effects of FO supplementation could be observed.

All HPNPs’ regimens were associated with higher ALP activities compared to those in HCs, although these were still within the normal range. An effect of added Omegaven was observed only in the Smoflipid regimen, where ALP activities decreased. The ALP enzyme is traditionally used as a marker of cholestasis and also as a surrogate marker of BA retention [37]. BAs are the main determinants of liver ALP activity and secretion. LPS also induces ALP activity, however, and may play a role in the anti-inflammatory response [38,39]. We also found a positive correlation of ALP values with the ratio of primary to secondary BAs, together with a trend toward a negative correlation with secondary BAs. Furthermore, there was a positive correlation between ALP and C4 values. These results suggest a link between possible low-grade inflammation and disruption of BA metabolism in HPNPs.

## 5. Conclusions

In conclusion, all HPN regimens are associated with increased SOD1 activities. The intervention of adding FO significantly influenced the spectrum of PPLs and reduced SOD1 values to the level of HCs, most likely because of decreased OS. The clinical relevance of this finding is not yet clear, however, and further studies are needed to clarify the influence of added FO on OS and antioxidant status in HPNPs.

## Figures and Tables

**Figure 1 nutrients-16-00485-f001:**
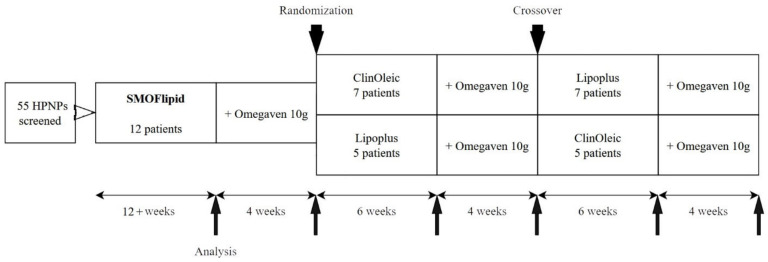
Study design. HPNPs, home parenteral nutrition patients.

**Figure 2 nutrients-16-00485-f002:**
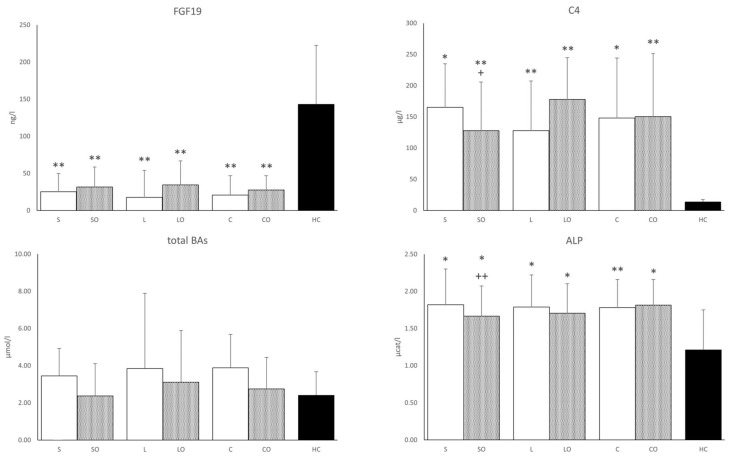
FGF19, cholesten, total bile acids, and alkaline phosphatase in serum. FGF19, fibroblast growth factor 19; C4, 7-α-hydroxy-4-cholesten-3-one; BAs, bile acids; ALP, alkaline phosphatase. Regimens (horizontal axis): S, Smoflipid; SO—Smoflipid+Omegaven; L, Lipoplus; LO, Lipoplus+Omegaven; C, ClinOleic; CO, ClinOleic+Omegaven. * HPN regimens vs. healthy controls (* *p* < 0.05, ** *p* < 0.01). FGF19, C4, and BAs were calculated using the Kruskal–Wallis test; ALP was calculated using one-way ANOVA; + Indicates Omegaven vs. baseline emulsion (+ *p* < 0.05, ++ *p* < 0.01). FGF19, C4, and BAs were calculated using the Wilcoxon signed-rank test; ALP was calculated using a paired sample *t*-test. Values of FGF19, C4, and BAs are median with a 75% percentile. Values of ALP are mean ± standard deviation (SD). Patients, *n* = 12; healthy controls, *n* = 12.

**Figure 3 nutrients-16-00485-f003:**
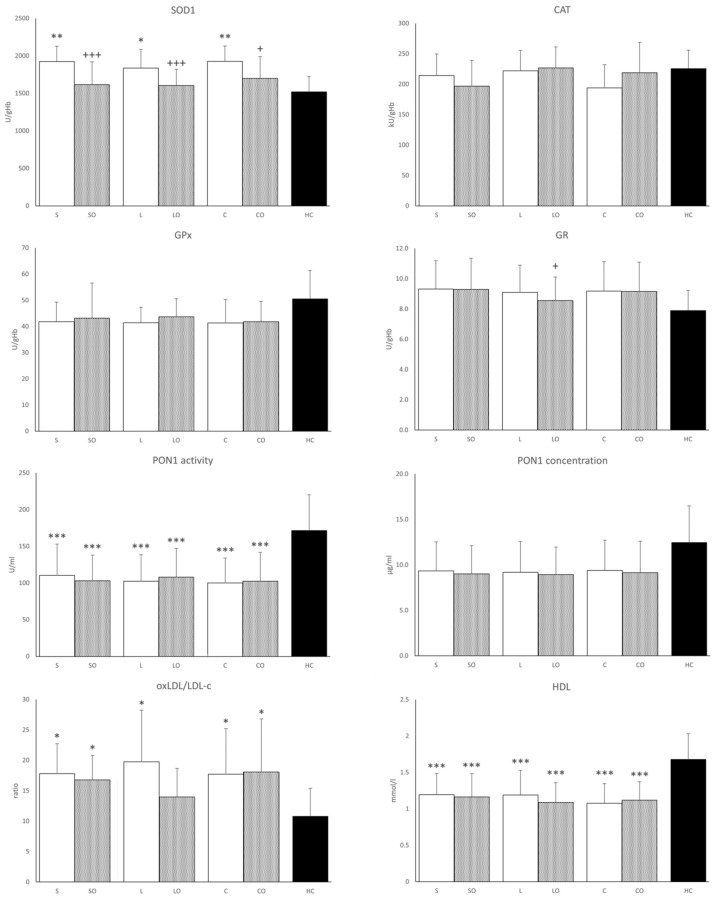
Antioxidant enzyme activities in serum, oxLDL/LDL cholesterol, and HDL cholesterol in serum. SOD1, Cu/Zn superoxide dismutase; CAT, catalase; GPx, glutathione peroxidase; GR, glutathione reductase; PON1, paraoxonase 1; oxLDL/LDL-c, oxidized low-density lipoprotein to low density lipoprotein-cholesterol ratio. Regimens (horizontal axis): S, Smoflipid; SO, Smoflipid+Omegaven; L, Lipoplus; LO, Lipoplus+Omegaven; C, ClinOleic; CO, ClinOleic+Omegaven. * Indicates HPN regimens vs. healthy control (* *p* < 0.05; ** *p* < 0.01; *** *p* < 0.001) were calculated using one-way ANOVA. + Omegaven vs. baseline emulsion (+ *p* < 0.05, +, +++ *p* < 0.001) were calculated using a paired sample *t*-test. Values are means ± standard deviation (SD). Patients, *n* = 12; healthy controls, *n* = 12.

**Table 1 nutrients-16-00485-t001:** Demographic parameters and classification of intestinal failure in home parenteral nutrition patients.

HPNPs								HCs		
Sex F/M	Age (years)	BMI	Infusions/Week	Infusion ^a^ (mL)	Energy ^b^ (kJ)	Enteral ^c^ (%)	Classification, IF ^d^ SBS/Other	Sex F/M	Age (years)	BMI
8/4	58.3 ± 17.6 ^e^	23.4 ± 3.7	5.8 ± 1.0	2910 ± 1200	72 ± 32	40 ± 16	10/2	8/4	61 ± 17	25.1 ± 3.0

BMI, body mass index; IF, intestinal failure; F, female; HCs, healthy controls; HPNPs, home parenteral nutrition patients; M, male; SBS, short bowel syndrome. Demographic parameters and classification of intestinal failure of individual patients were presented in a previous publication [2]. ^a^ Daily mean of total parenteral fluid volume (mL) infused per week. ^b^ Daily mean of total parenteral energy intake per week per kilogram of body weight. ^c^ Proportion of enteral nutrition in total energy intake. ^d^ ESPEN classification of intestinal failure in adults [1]. ^e^ Values are means ± standard deviation (SD).

**Table 2 nutrients-16-00485-t002:** Biochemical parameters in serum.

		Smoflipid				Lipoplus				ClinOleic				Healthy Controls
Parameter		Baseline	*p*	+Omegaven	*p*	Baseline	*p*	+Omegaven	*p*	Baseline	*p*	+Omegaven	*p*	
TAG	(mmol/L)	1.21 ± 0.44	§ 0.0297	1.21 ± 0.5		1.34 ± 0.6		1.54 ± 0.8		1.56 ± 0.6		1.43 ± 0.6		1.16 ± 0.5
TC	(mmol/L)	4.30 ± 1.0	* 0.0138	4.36 ± 1.0	* 0.0039	4.22 ± 1.0	* 0.0119	* 0.0039	* 0.0083	3.98 ± 1.0	* 0.0039	4.12 ± 0.8	* 0.0081	5.60 ± 1.1
LDL-C	(mmol/L)	2.57 ± 0.9		2.65 ± 1.0		2.43 ± 0.9		2.54 ± 1.0		2.20 ± 0.9		2.41 ± 0.8		3.4 ± 1.0
Apo A1	(g/L)	1.43 ± 0.3		1.28 ± 0.3	* 0.0002	1.29 ± 0.4	* 0.0002	1.26 ± 0.4	* 0.0002	1.31 ± 0.3	* 0.0001	1.19 ± 0.3	* 0.0001	1.67 ± 0.3
Apo B	(g/L)	0.91 ± 0.34		0.87 ± 0.35	* 0.0002	0.89 ± 0.33	* 0.0002	0.93 ± 0.41	* 0.0002	0.69 ± 0.24	* 0.0001	1.07 ± 0.46	* 0.0002	1.09 ± 0.39
Lp(a)	(g/L)	0.21 ± 0.21		0.21 ± 0.19		0.3 ± 0.29		0.28 ± 0.23		0.27 ± 0.25		0.2 ± 0.16		0.14 ± 0.27
SAA	(mg/L)	24.80 ± 15.6		33.57 ± 18.6		25.55 ± 20.4		27.98 ± 14.2		31.50 ± 21.0		34.12 ± 15.6		19.96 ± 24.3
HCY	(μmol/mL)	13.3 ± 3.4		14.2 ± 4.6		13 ± 4		13.6 ± 4.9		13.5 ± 2.8		14.8 ± 4.6		13 ± 2.5
Ca	(mmol/L)	2.24 ± 0.1	* 0.0285	2.20 ± 0.1	* 0.0103	2.24 ± 0.1	* 0.0372	2.24 ± 0.1	* 0.0228	2.22 ± 0.1	* 0.0221	2.27 ± 0.1	* 0.0437	2.37 ± 0.06
Fe	(μmol/L)	14.22 ± 7.47	§ 0.043	13.72 ± 5.32		15.09 ± 7.58		15.61 ± 6.5		17.89 ± 10.25		14.37 ± 6.5		22.2 ± 10.1
Transferrin	(mg/L)	3.47 ± 0.81	* 0.0204	3.45 ± 0.85	* 0.0102	3.46 ± 0.78	* 0.0122	3.5 ± 0.79	* 0.0148	3.31 ± 0.74	* 0.0201	3.25 ± 0.79	* 0.0122	2.45 ± 0.34
Ferritin	(μg/L)	151 ± 220.2		130.7 ± 167.9		145.7 ± 185.6		152.5 ± 185.3		152.8 ± 232.2		220.1 ± 223.4		94.5 ± 69
Hemoglobin	(g/L)	131 ± 11		130 ± 13		132 ± 11		131 ± 15		128 ± 14		131 ± 14		144 ± 9

TAG, triacylglycerols; TC, total cholesterol; LDL-C, low-density lipoprotein cholesterol; Apo A1, apolipoprotein A1; Apo B, apolipoprotein B; Lp(a), lipoprotein(a); SAA, serum amyloid A; HCY, homocysteine; Ca, calcium; Fe, iron; HPN, home parenteral nutrition. * HPN regimens vs. healthy control were calculated using one-way ANOVA with Newman–Keuls post hoc test comparisons; +Omegaven vs. baseline emulsion were calculated using paired sample *t*-test; § Smoflipid vs. ClinOleic regimen were calculated using one-way ANOVA. Values are means ± standard deviation (SD). Only significant *p* values are presented, meaning *p* < 0.05. Patients, *n* = 12; healthy controls, *n* = 12.

**Table 3 nutrients-16-00485-t003:** Serum bile acids and liver function tests.

		Smoflipid				Lipoplus				ClinOleic				Healthy Controls
		Baseline	*p*	+Omegaven	*p*	Baseline	*p*	+Omegaven	*p*	Baseline	*p*	+Omegaven	*p*	
Sum of primary BAs	(μmol/L)	3.96 [2.94; 4.9]		2.4 [2.14; 4.63]		4.42 [3.14; 6.63]		3.65 [2.39; 5.02]		4.18 [2.63; 6.33]		3.8 [2.4; 5]		1.48 [1.3; 3.25]
Sum of secondary BAs	(μmol/L)	0.45 [0.17; 0.94]		0.31 [0.17; 0.94]		0.55 [0.17; 1.44]		0.43 [0.17; 0.97]		0.38 [0.17; 0.92]		0.23 [0.17; 0.85]		0.65 [0.44; 1.17]
Primary/secondary BAs ratio		6.6 [2.9; 30.6]	* 0.033	6.6 [2.7; 32]	* 0.106	6.9 [2.7; 31.4]	* 0.054	9.2 [2.3; 53.1]	* 0.039	7.8 [3.2; 47.7]	* 0.007	9.1 [3; 31.7]	* 0.014	2.7 [1.5; 4.8]
Conjugated/unconjugated BAs ratio		13.6 [0.1;51.8]		7.4 [0.1; 48.4]		5.9 [0.1; 43.3]		12.4 [0.1; 91.6]		2.5 [0.3; 66]		7 [0.1; 59]		5.4 [0.9; 16.5]
ALT	(μcat/L)	0.61 ± 0.35		0.7 ± 0.44		0.89 ± 0.75		0.9 ± 0.81		0.91 ± 0.94		0.69 ± 0.43		0.36 ± 0.08
AST	(μcat/L)	0.48 ± 0.17		0.53 ± 0.19		0.58 ± 0.22		0.6 ± 0.27		0.61 ± 0.34		0.55 ± 0.14		0.41 ± 0.07
GGT	(μcat/L)	0.6 ± 0.54		0.65 ± 0.54		0.73 ± 0.57		0.74 ± 0.44		0.82 ± 0.72		0.83 ± 0.48		0.31 ± 0.19
Total bilirubin	(μmol/L)	13.3 ± 9.8		15.6 ± 13.2		17.9 ± 20.4		18.3 ± 17.7		19.4 ± 24.2		16.6 ± 14.3		10.4 ± 4.5
FGF19 vs. SOD1 ^c^		0.82	# 0.001	0.38	# 0.228	0.7	# 0.011	0.47	# 0.88	0.76	# 0.004	0.39	# 0.211	
ALP vs. C4 ^c^		0.43	# 0.166	0.47	# 0.122	0.79	# 0.002	0.58	# 0.047	0.72	# 0.008	0.71	# 0.009	
ALP vs. p/s BAs ^c^		0.65	# 0.021	0.57	# 0.053	0.51	# 0.088	0.64	# 0.026	0.69	# 0.012	0.63	# 0.028	

BAs, bile acids; p/s, primary to secondary BAs ratio; ALT, alanine transferase; AST, aspartate transferase; GGT, γ-glutamyltransferase; FGF19, fibroblast growth factor 19; SOD1, Cu/Zn superoxide dismutase; ALP, alkaline phosphatase; C4, 7-α-hydroxy-4-cholesten-3-one; HPN, home parenteral nutrition. * HPN regimens vs. healthy control were calculated using the Kruskal–Wallis test for BAs and one-way ANOVA for liver function tests; # two-tailed *p*-values. Bile acid values are expressed as the median and interquartile range. Level of significance: *p* < 0.05. Values of liver function tests are presented as means ± standard deviation (SD). ^c^ Spearman correlation coefficient. Patients, *n* = 12; healthy controls, *n* = 12.

**Table 4 nutrients-16-00485-t004:** Fatty acid patterns in plasma phospholipids.

	Smoflipid				Lipoplus				ClinOleic				Healthy Controls
Fatty Acids, Mol%	Baseline	*p*	+Omegaven	*p*	Baseline	*p*	+Omegaven	*p*	Baseline	*p*	+Omegaven	*p*	
16:00	31.40 ± 1.61	* 0.0002	31.52 ± 2.8	* 0.0002	30.61 ± 1.2	* 0.0022	30.27 ± 1.6	* 0.0061	30.27 ± 1.7	* 0.0033	29.97 ± 0.95	* 0.0043	27.89 ± 1.6
16:1*n*-7	0.61 ± 0.2		0.62 ± 0.2		0.64 ± 0.3		0.63 ± 0.3		0.65 ± 0.3		0.61 ± 0.2		0.59 ± 0.2
18:00	12.70 ± 1.7		12.92 ± 1.8		12.91 ± 1.4		13.28 ± 1.42		12.73 ± 1.2		12.83 ± 1.0		13.88 ± 1.2
18:1*n*-9	11.5 ± 1.2		11.28 ± 1.5		11.73 ± 2.0		11.42 ± 2.3		11.89 ± 1.7		12.65 ± 1.9	* 0.0286	10.38 ± 1.1
18:2*n*-6	16.67 ± 2.2	* 0.0006	15.13 ± 2.2	* 0.0001;	17.76 ± 3.3	* 0.0013	16.57 ± 3.6	* 0.0007	17.38 ± 3.1	* 0.0013	16.92 ± 3.0	* 0.0008	21.84 ± 3.1
				+ 0.033				+ 0.0356					
18:3*n*-3	0.16 ± 0.04	* 0.0199	0.15 ± 0.1	* 0.0095	0.16 ± 0.04	* 0.0284	0.17 ± 0.1	* 0.0369	0.18 ± 0.06	* 0.0391	0.14 ± 0.01	* 0.0047	0.22 ± 0.04
20:3*n*-6	2.75 ± 1.1	§ 0.0093	2.08 ± 1.0	* 0.0327	2.83 ± 1.1		2.14 ± 0.8	* 0.0344	3.69 ± 1.3		2.16 ± 0.8	* 0.0261	3.284 ± 0.6
				+ 0.0083				+ 0.0191				+ 0.0004	
20:4*n*-6	8.66 ± 2.0	* 0.0002	7.20 ± 1.7	* 0.0001	8.36 ± 1.5	* 0.0002	7.48 ± 1.6	* 0.0001	9.57 ± 2.1	* 0.0016	7.64 ± 1.6	* 0.0001	11.89 ± 1.4
				+ 0.00001				+ 0.0002				+ 0.0004	
20:5*n*-3	3.34 ± 1.0	* 0.0021	5.18 ± 1.8	* 0.0001	3.08 ± 1.5	* 0.0041	4.96 ± 2.07	* 0.0001	2.18 ± 1.0		4.55 ± 2.0	* 0.0001	1.1 ± 0.4
		§ 0.0042		+ 0.0002				+ 0.0001				+ 0.0003	
22:5*n*-3	1.52 ± 0.3	* 0.0037	1.66 ± 0.4	* 0.0003	1.475 ± 0.3	* 0.0072	1.53 ± 0.4	* 0.0044	1.37 ± 0.3	* 0.0342	1.26 ± 0.23	+ 0.0043	1.038 ± 0.07
22:6*n*-3	6.98 ± 1.3	* 0.0002	8.70 ± 2.9	* 0.0001	6.68 ± 1.4	* 0.0002	7.93 ± 1.5	* 0.0001	6.09 ± 1.9	* 0.0009	7.4 ± 1.5	* 0.0001	3.58 ± 1.0
								+ 0.0001				+ 0.001	
Ʃ SFA	45.22 ± 1.4	* 0.007	45.66 ± 2.6	* 0.0012	44.43 ± 2.1		44.48 ± 1.6		43.89 ± 1.2		43.69 ± 0.8		42.83 ± 0.9
Ʃ MUFA	14.19 ± 1.38		13.91 ± 1.62		14.45 ± 2.46		14.11 ± 2.54		14.74 ± 1.88		15.47 ± 1.92	* 0.0322	12.96 ± 1.2
Ʃ *n*-6 PUFA	28.81 ± 2.2	* 0.0001	25.13 ± 3.4	* 0.0001	29.81 ± 3.2	* 0.0001	26.92 ± 3.6	* 0.0001	31.61 ± 3.0	* 0.0001	27.53 ± 3.7	* 0.0001	38.16 ± 2.2
		§ 0.0025		+ 0.0001				+ 0.00001				+ 0.0001	
Ʃ *n*-3 PUFA	11.78 ± 2.2	* 0.0002	15.3 ± 4.2	* 0.0001	11.31 ± 2.1	* 0.0002	14.49 ± 2.9	* 0.0001	9.76 ± 2.8	* 0.0019	13.31 ± 3.1	* 0.0001	6.05 ± 1.4
		§ 0.0082		+ 0.0062				+ 0.00001				+ 0.00001	
*n*-6 + *n*-3	40.58 ± 2.0	* 0.0028	40.43 ± 2.9	* 0.0022	41.11 ± 2.5	* 0.0073	41.41 ± 2.6	* 0.0036	41.37 ± 2.2	* 0.0088	40.84 ± 2.1	* 0.0046	44.21 ± 1.4
SFA+MUFA/PUFA	1.46 ± 0.12	* 0.0058	1.48 ± 0.17	* 0.0034	1.42 ± 0.13	* 0.0124	1.45 ± 0.13	* 0.0153	1.44 ± 0.14	* 0.0058	1.42 ± 0.15	* 0.0089	1.26 ± 0.07
20:4/20:5	2.88 ± 1.2	* 0.0001	1.57 ± 0.7	* 0.0001	3.35 ± 1.6	* 0.0001	1.71 ± 0.6	* 0.0001	5.44 ± 2.8	* 0.0001	2.03 ± 1.0	* 0.0001	3.56 ± 1.4
		§ 0.0052		+ 0.0001				+ 0.0016				+ 0.0006	
20:4/22:6	1.27 ± 0.4	* 0.0001	0.9 ± 0.3	* 0.0001	1.3 ± 0.4	* 0.0001	0.96 ± 0.2	* 0.0001	1.77 ± 0.9	* 0.0001	1.07 ± 0.3	* 0.0001	3.52 ± 0.9
		§ 0.0149		+ 0.0001				+ 0.0002				+ 0.0028	
22:6/18:3	31.42 ± 15.1	* 0.0101	44.98 ± 24.0	* 0.0001	29.99 ± 16.6	* 0.0105	35.74 ± 16.3	* 0.0027	22.73 ± 12.7		34.01 ± 10.0	* 0.0045	11.46 ± 3.6
		§ 0.0009										+ 0.0005	

SFA, saturated fatty acids (FA); MUFA, monounsaturated FA; *n*-6 PUFA, *n*-6 polyunsaturated FA; *n*-3 PUFA, *n*-3 polyunsaturated FA. * HPN regimens vs. healthy control were calculated using one-way ANOVA with the Newman–Keuls post hoc test; + Omegaven vs. baseline emulsion were calculated using a paired sample *t*-test. § Smoflipid vs. ClinOleic regimen were calculated using one-way ANOVA. Only significant *p* values are presented, *p* < 0.05 is significant. Values are means ± standard deviation (SD). Patients, *n* = 12; healthy controls, *n* = 12.

## Data Availability

Data are contained within the article.

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
