# Peer review of "High Dose Fish Oil Added to Various Lipid Emulsions Normalizes Superoxide Dismutase 1 Activity in Home Parenteral Nutrition Patients"

_nutrients, 2024, doi:10.3390/nu16040485_

Round 1

Reviewer 1 Report

Comments and Suggestions for Authors

Fish oil emulsion Review

1.       What is the main question addressed by the research?

This study investigated how high dose of fish oil added to various 16 lipid emulsions influences antioxidant status and liver function markers in HPNPs.

The manuscript needs many more details than are presented. For example, I learned from Google “Each mL of Omegaven contains 0.1 g of fish oil, 0.012 g egg phospholipids, 0.025 g glycerin, 0.15 to 0.3 mg dl-alpha-tocopherol, 0.3 mg sodium oleate, water for injection, and sodium hydroxide for pH adjustment (pH 6 to 9).” Such information should be provided, perhaps in a table, for the reader to understand the differences in emulsions in detail.

2.       What parts do you consider original or relevant for the field? What
specific gap in the field does the paper address?

3.      What does it add to the subject area compared with other published
material?

The authors have not measured vitamin E, but the fat malabsorption caused by short bowel syndrome, the poor bile acid recirculation and the generally low vitamin E intake makes me think that low vitamin E status is likely. (See for example a case report: Howard L, et al. Reversible neurological symptoms caused by vitamin E deficiency in a patient with short bowel syndrome. Am J Clin Nutr. 1982;36(6):1243-9.) Note too that vitamin E can decrease SOD (Kır HM, et al. Effects of Vitamins E, A and D on MDA, GSH, NO Levels and SOD Activities in 5/6 Nephrectomized Rats. American Journal of Nephrology. 2005;25(5):441-6. doi: 10.1159/000087825.)

Additionally, FGF decreased in rats with diabetes given antioxidants, including vitamin E. (da Purificação NRC, et al. Combined use of systemic quercetin, glutamine and alpha-tocopherol attenuates myocardial fibrosis in diabetic rats. Biomed Pharmacother. 2022 Jul;151:113131. doi: 10.1016/j.biopha.2022.113131. PMID: 35643067.)

5. Please describe how the conclusions are or are not consistent with the
evidence and arguments presented. Please also indicate if all main questions
posed were addressed and by which specific experiments.
6. Are the references appropriate?

Taken together the information known about vitamin E suggests that the Omegaven may be beneficial not only because it contains fish oil, but also because it contains alpha-tocopherol.  It would be useful to report the circulating alpha-tocopherol in these subjects at each time point. At least a discussion of the benefit of vitamin E in decreasing oxidative stress should be discussed. It is generally recognized that short bowel syndrome results in vitamin E deficiency that can ultimately result in neurologic abnormalities, which cannot be reversed (e.g. peripheral neuropathy resulting in ataxia). It is suggested that this complication be discussed so that physicians treating such patients are reminded of the importance of vitamin E supplementation, especially since these new lipid emulsions provide a means to reverse the deficiency.

4. What specific improvements should the authors consider regarding the
methodology? What further controls should be considered?

Specific comments

Abstract: “Smoflipid for at least 3 months were given fish oil (Omegaven) for a further 4 weeks. Then, the patients were randomized to receive subsequently Lipoplus and ClinOleic for 6 weeks or vice versa plus 4 weeks of Omegaven after each cycle in a crossover design.” The reader is at a loss at to what these various treatments are and how they differ. Information detailing the major differences should be provided in the abstract. Possibly abbreviations could be used for the various emulsions so there is sufficient room for other details, e.g. Which contain fish oil, etc. The following statement from the methods would be useful in the abstract (lines98-102)

There are a number of undefined abbreviations in the abstract, which also need clarification, e.g. FGF19.

Introduction

“This study follows on our previously published clinical trial comparing two FO-enriched LEs, Smoflipid with natural FO and Lipoplus with re-esterified FO, with ClinOleic, an LE with low PUFAs and high MUFAs [2]. Using Omegaven, this study tested additional supplementation of FO in all three baseline LEs and demonstrated the suppression of in vitro LPS-stimulated cytokine production by escalated dose of FO.” This sentence starts to define the lipid emulsions but fails to define the kinds of lipids. These abbreviations all must be defined.

Methods.

Lines 102-3 “After at least 12 weeks of Smoflipid, Omegaven™ (containing 10% FO) was added for a further 4 weeks.” The diagram does not show any period of Smoflipid given for 12 weeks…only 6 weeks. Please clarify this discrepancy

Line 105 How many hours after the last emulsion administration was the blood sample taken?

7. Please include any additional comments on the tables and figures and
quality of the data.

Table 2. What statistical tests were used? The results paragraph described comparison to HC group, but it is unclear how the comparisons were made. There also seems to be a within group comparison to test the effects of omegaven.

Table 3 same comments as for Table 2

Comments on the Quality of English Language

Acceptable after the definitions of abbreviations are provided.

Author Response

We thank for the comments, which, as we believe, have substantially improved our paper.
Points raised are in italics.

Our responses are in normal text.

  1. Minor editing of English language was required. 

R: We improved the manuscript for language quality by English Editing Services (www.englisheditingservices.com). 

  1. The manuscript needs many more details than are presented. For example, I learned from Google “Each mL of Omegaven contains 0.1 g of fish oil, 0.012 g egg phospholipids, 0.025 g glycerin, 0.15 to 0.3 mg dl-alpha-tocopherol, 0.3 mg sodium oleate, water for injection, and sodium hydroxide for pH adjustment (pH 6 to 9).” Such information should be provided, perhaps in a table, for the reader to understand the differences in emulsions in detail.

R: We agree with reviewer. We have already presented the detailed lipid emulsion contents in the previous sub-study article ref. [2]. We added following statements into manuscript. In methods: “All formula components including daily vitamin and trace element supplements were used in accordance with the European Summary of Product Characteristics. LE compositions of individual oils, egg phospholipids, vitamin E (all patients were monitored at least twice a year to maintain normal levels of vitamin E) and FO dosing, …”.

  1. The authors have not measured vitamin E, but the fat malabsorption caused by short bowel syndrome, the poor bile acid recirculation and the generally low vitamin E intake makes me think that low vitamin E status is likely. (See for example a case report: Howard L, et al. Reversible neurological symptoms caused by vitamin E deficiency in a patient with short bowel syndrome. Am J Clin Nutr. 1982;36(6):1243-9.) Note too that vitamin E can decrease SOD (Kır HM, et al. Effects of Vitamins E, A and D on MDA, GSH, NO Levels and SOD Activities in 5/6 Nephrectomized Rats. American Journal of Nephrology. 2005;25(5):441-6. doi: 10.1159/000087825.)

R: The patients were certainly not in tocopherol deficiency. All patients were regularly monitored twice a year for vitamin E status and supplemented accordingly. Daily vitamin and trace elements were part of all parenteral bags. Therefore patients were prone to receive sufficient doses of alpha tocopherol in fish oil. As for Clinoleic, it also contained other tocopherols, please see ref. [2]. We added this to the text in methodology section, please see response no.2. 

  1. Taken together the information known about vitamin E suggests that the Omegaven may be beneficial not only because it contains fish oil, but also because it contains alpha-tocopherol.  It would be useful to report the circulating alpha-tocopherol in these subjects at each time point. At least a discussion of the benefit of vitamin E in decreasing oxidative stress should be discussed. It is generally recognized that short bowel syndrome results in vitamin E deficiency that can ultimately result in neurologic abnormalities, which cannot be reversed (e.g. peripheral neuropathy resulting in ataxia). It is suggested that this complication be discussed so that physicians treating such patients are reminded of the importance of vitamin E supplementation, especially since these new lipid emulsions provide a means to reverse the deficiency.

R: We fully agree, but it is rather difficult to discuss the role of vitamin E in oxidative stress. We added following statement into manuscript. In discussion: “However, an antioxidant effect of alpha-tocopherol, which was given to the patients together with a high dose of fish oil, should also be considered. There are studies that confirm the protective effect of this vitamin against the peroxidation of PUFAs, but in general the results are quite controversial, many studies also demonstrate its pro-oxidant effect [19].”

  1. Abstract: “Smoflipid for at least 3 months were given fish oil (Omegaven) for a further 4 weeks. Then, the patients were randomized to receive subsequently Lipoplus and ClinOleic for 6 weeks or vice versa plus 4 weeks of Omegaven after each cycle in a crossover design.” The reader is at a loss at to what these various treatments are and how they differ. Information detailing the major differences should be provided in the abstract. Possibly abbreviations could be used for the various emulsions so there is sufficient room for other details, e.g. Which contain fish oil, etc. The following statement from the methods would be useful in the abstract (lines98-102) There are a number of undefined abbreviations in the abstract, which also need clarification, e.g. FGF19.

R: We edited the abstract according to the reviewer's requirements. We have added explanations for abbreviations. Since the abstract is limited to 200 words, it is not possible to supplement the proposed text from the methodology line 98-102 (70 words). Due to the detailed description in the methodology, including the diagram (figure 1), we believe that the reader will get the necessary details of the study.

  1. Introduction “This study follows on our previously published clinical trial comparing two FO-enriched LEs, Smoflipid with natural FO and Lipoplus with re-esterified FO, with ClinOleic, an LE with low PUFAs and high MUFAs [2]. Using Omegaven, this study tested additional supplementation of FO in all three baseline LEs and demonstrated the suppression of in vitro LPS-stimulated cytokine production by escalated dose of FO.” This sentence starts to define the lipid emulsions but fails to define the kinds of lipids. These abbreviations all must be defined.

R: We carefully revised the abbreviations, all are defined in abstract and the whole manuscript. 

  1. Methods. Lines 102-3 “After at least 12 weeks of Smoflipid, Omegaven™ (containing 10% FO) was added for a further 4 weeks.” The diagram does not show any period of Smoflipid given for 12 weeks…only 6 weeks. Please clarify this discrepancy

R: We fixed the error in the scheme. Thank you.

  1. Line 105 How many hours after the last emulsion administration was the blood sample taken?

R: At least 5 hours after finishing the parenteral nutrition administration, we added this to the text

  1. Table 2. What statistical tests were used? The results paragraph described comparison to HC group, but it is unclear how the comparisons were made. There also seems to be a within group comparison to test the effects of omegaven.Table 3 same comments as for Table 2

R: We added the statistical methods into the table and figure legends.

  1. Acceptable after the definitions of abbreviations are provided.

R: We defined and checked all definitions of abbreviations in abstract and in the whole manuscript.

Thank you for your review and valuable suggestions.

Reviewer 2 Report

Comments and Suggestions for Authors

This study investigated how high dose of fish oil added to various lipid emulsions influences antioxidant status and liver function markers in HPNPs. Twelve HPNPs receiving Smoflipid for at least 3 months were given fish oil (Omegaven) for a further 4 weeks. Then, the patients were randomized to receive subsequently Lipoplus and ClinOleic for 6 weeks or vice versa plus 4 weeks of Omegaven after each cycle in a crossover design. Twelve age- and sex-matched healthy controls (HCs) were included. Results: Superoxide dismutase (SOD1) activity and oxidized-low density lipoprotein concentration were higher in all baseline HPN regimens compared to HCs. The Omegaven lowered SOD1 compared to baseline regimens and thus normalized it toward HCs. Lower paraoxonase 1 activity and FGF19 concentration, conversely higher ALP activity and cholesten (C4) concentration were observed in all baseline regimens compared to HCs. Close correlation was observed between FGF19 and SOD1 in all baseline regimens. Conclusion: Escalated dose of fish oil normalized SOD1 activity in HPNPs toward that of HCs. BA metabolism was altered in HPNPs without signs of significant cholestasis, which was not affected by Omegaven.

It is interesting topic and the result is positive. However, there still have some issue need to check.

1.       The font format needs to be unified.

2.      “3.2. Biochemical Parameters” There should be analyzed in scientific difference.

3.       “3.4. Antioxidant Enzymes”. The reason for the oxidative stress test should be included. Please refer this reference (Food & Function, 2022, 13(24), 12686-12696.).

4.      The Line number for discussion should not be separated. The antiinflammation for n-3 PUFAs should refer this reference (Cancer Research, 2020, 80(12): 2564-2574.).

5.      The reference should be updated in recent years.

Comments on the Quality of English Language

This study investigated how high dose of fish oil added to various lipid emulsions influences antioxidant status and liver function markers in HPNPs. Twelve HPNPs receiving Smoflipid for at least 3 months were given fish oil (Omegaven) for a further 4 weeks. Then, the patients were randomized to receive subsequently Lipoplus and ClinOleic for 6 weeks or vice versa plus 4 weeks of Omegaven after each cycle in a crossover design. Twelve age- and sex-matched healthy controls (HCs) were included. Results: Superoxide dismutase (SOD1) activity and oxidized-low density lipoprotein concentration were higher in all baseline HPN regimens compared to HCs. The Omegaven lowered SOD1 compared to baseline regimens and thus normalized it toward HCs. Lower paraoxonase 1 activity and FGF19 concentration, conversely higher ALP activity and cholesten (C4) concentration were observed in all baseline regimens compared to HCs. Close correlation was observed between FGF19 and SOD1 in all baseline regimens. Conclusion: Escalated dose of fish oil normalized SOD1 activity in HPNPs toward that of HCs. BA metabolism was altered in HPNPs without signs of significant cholestasis, which was not affected by Omegaven.

It is interesting topic and the result is positive. However, there still have some issue need to check.

1.       The font format needs to be unified.

2.      “3.2. Biochemical Parameters” There should be analyzed in scientific difference.

3.       “3.4. Antioxidant Enzymes”. The reason for the oxidative stress test should be included. Please refer this reference (Food & Function, 2022, 13(24), 12686-12696.).

4.      The Line number for discussion should not be separated. The antiinflammation for n-3 PUFAs should refer this reference (Cancer Research, 2020, 80(12): 2564-2574.).

5.      The reference should be updated in recent years.

Author Response

We thank for the comments, which, as we believe, have substantially improved our paper.
Points raised are in italics.

Our responses are in normal text.

  1. Moderate editing of English language was required.

R: We improved the manuscript for language quality by English Editing Services (www.englisheditingservices.com). 

  1. The font format needs to be unified.

R: The font format was unified to times new roman.

  1. “3.2. Biochemical Parameters” There should be analyzed in scientific difference.

R: We fixed the description in the tables and added the details of statistical evaluation.

  1. “3.4. Antioxidant Enzymes”. The reason for the oxidative stress test should be included. Please refer this reference (Food & Function, 2022, 13(24), 12686-12696.).

R: The association of intestinal failure, inflammation and oxidative stress is mentioned in introduction section of the manuscript. We added suggested reference into introduction as no. [15].

  1. The Line number for discussion should not be separated. The antiinflammation for n-3 PUFAs should refer this reference (Cancer Research, 2020, 80(12): 2564-2574.).

R: We added suggested reference into discussion as no. [28]. The line numbers are assigned by Editorial manager automatically. We cannot influence the process.

  1. The reference should be updated in recent years.

R: We have updated the references as suggested.

Thank you for your review and valuable suggestions.

Round 2

Reviewer 2 Report

Comments and Suggestions for Authors

It can be accepted in the current revision.

Comments on the Quality of English Language

It can be accepted in the current revision.